# Use of Phosphorus-Solubilizing Microorganisms as a Biotechnological Alternative: A Review

**DOI:** 10.3390/microorganisms12081591

**Published:** 2024-08-05

**Authors:** Efrén Venancio Ramos Cabrera, Zuly Yuliana Delgado Espinosa, Andrés Felipe Solis Pino

**Affiliations:** 1Escuela de Ciencias Agrícolas, Pecuarias y del Medio Ambiente—ECAPMA, Universidad Nacional Abierta y a Distancia—UNAD, Calle 5 # 46N-67, Popayán 190001, Cauca, Colombia; efren.ramos@unad.edu.co; 2Facultad de Ingeniería, Corporación Universitaria Comfacauca—Unicomfacauca, Cl. 4 N. 8-30, Popayán 190001, Cauca, Colombia; zdelgado@unicomfacauca.edu.co; 3Facultad de Ingeniería Electrónica y Telecomunicaciones, Universidad del Cauca, Popayán 190003, Cauca, Colombia

**Keywords:** phosphorus-solubilizing microorganisms, biofertilizers, sustainable agriculture, bibliometric analysis

## Abstract

Microorganisms with the ability to dissolve phosphorus have the potential to release this essential nutrient into the soil through natural solubilization processes, which allows for boosting plant growth and development. While literature reviews acknowledge their potential, unexplored territories concerning accessibility, application, and effective integration into sustainable agriculture necessitate further research. This manuscript employed distinct methodologies to execute a bibliometric analysis and a literature review. The combined application of both methodologies enables a holistic understanding of the domain landscape and its innovative facets. For the bibliometric analysis, the propositions of Donthu and Jia were utilized, supplemented by tools, such as Bibliometrix. The literature review adhered to a systematic methodology predicated on Petersen’s guidelines to represent the domain accurately, pinpointing trends and gaps that could steer future, more detailed research. This investigation uncovers an escalating interest in studying these microorganisms since the 2000s, emphasizing their significance in sustainable agriculture and the context of phosphorus scarcity. It was also discerned that India and China, nations with notable agricultural sectors and a high demand for phosphorus fertilizers, spearheaded research output on this subject. This signifies their substantial contribution to the progression of this scientific field. Furthermore, according to the research consulted, phosphorus-solubilizing microorganisms play a pivotal role in the symbiotic interaction of soil with plant roots and represent an efficacious strategy to counteract the low availability of phosphorus in the soil and sustainably enhance agricultural systems. Finally, this review contributes to the relevant domain by examining existing empirical evidence with special emphasis on sustainable agriculture, improved understanding of phosphorus solubilization mechanisms, and recognition of various microbial entities.

## 1. Introduction

Phosphorus is an essential nutrient for plant growth and development, playing a crucial role in plant physiology and reproduction. It is present in the fundamental molecules of life, such as nucleic acids, phospholipids, Adenosine Triphosphate (ATP), and other indispensable biological substances [1,2]. Plants’ demand for phosphorus is the second highest after nitrogen, limiting agricultural productivity and production [3]. Although soil may contain a significant amount of phosphorus, its uptake is restricted by soil chemical conditions, such as pH, and high concentrations of elements, such as aluminum and iron [4]. Phosphorus is highly reactive and soluble in a narrow pH range (neutral to slightly acidic) [5]. In acid soils, phosphorus complexes with aluminum and iron, whereas in alkaline soils, it complexes with calcium and magnesium to form insoluble phosphate compounds that plants cannot assimilate due to their cation and ammonium exchange capacity [6]. This limitation represents one of the main challenges for agricultural production, as it directly impacts crop yield [3].

Fertilizers are fundamental to increasing agricultural productivity by providing plants with the essential nutrients necessary for their growth and development [7]. These can be of natural origin, chemically synthesized, organic, or inorganic and contain nitrogen, phosphorus, potassium, calcium, sulfur, and magnesium [8]. Fertilizers help improve soil fertility, increase crop yields, improve product quality, and prevent plant nutrient deficiencies [9]. Globally, it is estimated that about 30 million tons of phosphate fertilizers are used to address phosphorus deficiency and improve crop productivity [10]. However, up to 80% of the phosphorus in these fertilizers is lost due to immobilization in soil colloids, preventing its conversion into plant-assimilable organic forms [11,12].

In recent years, extensive research has been conducted to address this problem by studying the excessive use of chemically synthesized phosphorus fertilizers, which often contain heavy metals, such as cadmium and lead [13]. In addition, these fertilizers contain strong acids, such as nitric acid, sulfuric acid, and hydrochloric acid, which can accumulate in the soil and reduce the beneficial microbial population, increasing soil acidity and ultimately affecting plant growth [14]. Therefore, biotechnological alternatives are currently being investigated, such as the search for microorganisms capable of solubilizing phosphate present in the soil into insoluble forms, which can improve the sustainability of agriculture by avoiding the need to expand the use of phosphorus chemical fertilizers that pollute the environment and cause effects, such as eutrophication or risks to human health [15].

Various studies and reviews have focused on understanding the role of phosphorus-solubilizing microorganisms and ways to improve their effectiveness in specific contexts. For instance, Kour in [16] highlights the significance of phosphorus as a crucial macronutrient for plant development and its limited mobility in plants and soil, affecting plant growth. The use of phosphorus solubilizers in combination with other plant-growth-promoting microbes is suggested to enhance phosphorus uptake in crops. Another perspective in [17] discusses the adverse effects of conventional phosphorus fertilizers and proposes inoculating seeds, crops, and soils with phosphate-solubilizing bacteria (PSB) as a promising strategy to improve food production without harming the environment. This review explores various soil microorganisms capable of solubilizing phosphorus and their potential as biofertilizers. Additionally, Sattar et al. [18] emphasize the importance of potassium as an essential nutrient for plants and its role in various physiological and metabolic processes.

Along this line, intensive cultivation practices have led to the depletion of soil fertility, particularly phosphorus, affecting the balance and diversity of soil microorganisms. Many of these microorganisms are phosphorus solubilizers that utilize organic acid production, pH lowering, acidolysis, chelation, and exchange reactions to convert unavailable phosphorus into accessible forms. Consequently, several phosphorus-solubilizing microorganisms are being marketed as biofertilizers to reduce reliance on chemical fertilizers and better use phosphorus reserves in soils.

Considering those described above, this investigation distinguishes itself from analogous inquiries by amalgamating two disparate yet complementary methodologies: bibliometric analysis and a literature review. This fusion provides a comprehensive viewpoint on the current state of phosphorus-solubilizing microorganisms as a biotechnological strategy for augmenting phosphorus accessibility. The extensive temporal range of the scrutinized studies, focused on revealing underlying mechanisms and delineating future research pathways, sets this manuscript apart from its contemporaries. Consequently, this endeavor furnishes substantial and meaningful advancements toward the progression and understanding of this scientific discipline.

The novelty of this study lies in its comprehensive approach, combining bibliometric analysis with an in-depth literature review to provide a holistic understanding of phosphorus-solubilizing microorganisms as a biotechnological strategy. This dual methodology allows for a quantitative assessment of research trends and a qualitative exploration of the mechanisms and applications of these microorganisms. The motivation for this work stems from the urgent need to address global phosphorus deficiency in agriculture while minimizing environmental impact. This study aims to accelerate the development and implementation of sustainable phosphorus management strategies in agriculture by synthesizing current knowledge and identifying research gaps.

The literature review revealed that similar systematic reviews or mappings do not supplement their reviews with bibliometric scrutiny. This oversight engenders a noticeable void in our comprehension, originating from the need for more application of statistical and mathematical methodologies to examine and construct metrics related to the dynamic and evolutionary patterns of scientific and technological data within the domain of solubilizing bacteria. The current investigation aims to bridge this knowledge gap by undertaking such an analysis.

## 2. Review Execution

This section of the paper outlines the systematic planning and execution of a bibliometric analysis and literature review focused on utilizing solubilizing microorganisms to enhance plant phosphorus availability. The bibliometric analysis primarily examines the authors, sources, and relevant countries associated with this subject [19]. Conversely, the literature review focuses on microorganisms and their role in improving plant phosphorus availability. By employing these techniques, a precise and methodical representation of the current state of research in this field is achieved. The analysis also helps to identify trends, areas of interest, and gaps that warrant further investigation [20].

The approach proposed by Donthu in [20] was adopted as the primary reference for the bibliometric analysis. Donthu’s proposal provides essential guidelines and instructions for conducting such studies and follows the Science Mapping Workflow methodology, which facilitates data acquisition, analysis, and visualization [21]. Additionally, the study incorporates the insights from Jia’s paper in [22], serving as a foundation for analyzing solubilizing microorganisms.

The literature review plan was executed systematically and reproducibly, ensuring organized information collection. This review adhered to the methodological process of Petersen’s guidelines in [23]. Several essential tools were employed to carry out this research. R-studio, which integrates the Bibliometrix library [24], was used to facilitate the execution of the bibliometric study. VOSviewer [25] was used to explore co-occurrence networks, employing text mining techniques to examine various inter-domain relationships. Finally, Google Sheets [25] and R-studio were used to generate the research graphs.

### 2.1. Information Collection

A search strategy was applied to several bibliographic databases to collect relevant information for the bibliometric analysis and literature review. The search strategy, outlined in Table 1, was constructed based on the PICOC criteria [26]. These criteria ensure that the search string logically and methodically covers all topics in the research scope. For this study, data were extracted from databases, such as Scopus, Web of Science, Dimensions AI, and ScienceDirect, and a manual search was performed using Google Scholar. These sources are widely accepted for searching the scientific literature in engineering and other research areas [27].

The search string used in this study was divided into several sections to narrow the search within the domain of solubilizing bacteria and their role in enhancing phosphorus assimilation in plants. The initial part of the search focused on phosphorus as the research’s central element and critical point. The following section examined fertilizers and their application to improve plant conditions. The third part of the search chain specified the specific area of interest, namely agriculture and cultivation. Finally, the fourth stage investigated plants’ desired action of phosphorus assimilation.

### 2.2. Definition of the Research Questions

The research questions posed for the systematic literature review seek to understand in what context phosphorus-solubilizing microorganisms are being used to improve the ability of plants to assimilate this mineral. This makes it possible to find the existing deficiencies in the research area and how to improve this process. This is intended to identify gaps or deficiencies in the research area and propose methods to improve this process. Table 2 outlines the individual research questions and their underlying motivations.

### 2.3. Methodology and Selection of Primary Studies

A methodological and systematic approach was adopted to ensure a rigorous and structured literature review, following some of the guidelines proposed by Petersen in [23], as they provide a clear framework for defining the scope of the review and, consequently, the primary studies. Figure 1 illustrates the information collection process, allowing us to address the studies considered effectively.

### 2.4. Execution of the Search

Research studies were selected with precision and systematic rigor through a tripartite methodology. A bespoke search query was formulated and applied to the titles, abstracts, and keywords across many databases in the initial phase. This preliminary search yielded 2322 scholarly articles, from which articles and conference papers were retained while content deemed irrelevant was excluded. A manual selection was also performed using Google Scholar to refine the selection, resulting in a subset of pertinent articles. This subset was subsequently reduced to 186 “candidate primary studies” based on examining their titles and abstracts. In the second phase, a similar methodology was employed, but with the acquisition and comprehensive reading of the full texts of the articles. This reading approach facilitated the identification of 155 primary articles that satisfied the inclusion criteria. The final phase involved systematically extracting relevant information from these 155 primary articles to address the research questions. An online tool named Parsifal [26], engineered explicitly for systematic reviews across various domains, was utilized to streamline and expedite the systematic review process.

## 3. Results and Analysis of Results

This section presents the domain’s primary contributions and data analysis while comprehensively describing and examining the gathered information. The bibliometric analysis encompasses descriptive and inferential techniques applied to the articles under study. It is crucial to emphasize that the bibliometric analysis relies exclusively on data from Scopus and Web of Science sources, owing to the constraints imposed by the Bibliometrix library, which necessitates specific metadata for domain analysis [19]. Regrettably, other databases cannot be utilized for this purpose due to their inability to furnish the requisite metadata. Furthermore, the latter part of this section furnishes responses to the posed inquiries aimed at addressing specific gaps in the existing literature, as identified during the literature review.

### 3.1. Execution of the Bibliometric Analysis

#### 3.1.1. Domain Overview

A bibliometric analysis was performed to assess the utilization of solubilizing microorganisms as a biotechnological method for enhancing the availability of assimilable phosphorus to plants. The studies spanned from 1982 to 2023, totaling 41 years of research data, indicating the maturation of this domain. A comprehensive examination of 495 sources, including journals, books, and other pertinent documents, within the Scopus and Web of Science databases identified 1168 relevant publications, underscoring the extensive literature on this subject.

The analysis unveiled an annual growth rate of 9.31% in the number of documents related to solubilizing microorganisms (Figure 2). This percentage signifies a consistent increase in research activity over time. The papers’ average age was 5.89 years, indicating that research in this field is relatively recent. Moreover, each paper received an average of 26.48 citations, highlighting the research’s influential nature and impact on the scientific community.

Regarding the authors in this domain, the analysis identified contributions from 4253 individuals who have participated in the research, highlighting the collaborative approach within the field. Of the total number of articles analyzed, 32 had a single author, indicating that most articles involved multiple authors, reflecting collaborative efforts in the domain. On average, each article had 4.9 co-authors, further illustrating the collaborative nature of the research. International co-authorships accounted for 14.3% of the total collaborations, suggesting the involvement of researchers from different countries in the study of phosphorus-solubilizing microorganisms as a biotechnological alternative [28]. Finally, the most common types of documents in this domain are research articles (937) and book chapters (87), indicating a preference for these formats when disseminating research results.

Overall, the data indicate a growing interest in the study of solubilizing microorganisms over time. A strong emphasis on collaborative research has led to a substantial body of literature with significant citation impact.

#### 3.1.2. Annual Scientific Production

The scientific output related to phosphorus-solubilizing microorganisms (Figure 3) has grown substantially in recent years, signifying a heightened interest and activity within the scientific community concerning this field. Initially, during the early 1980s, when the field was still emerging and establishing its fundamentals, the number of published articles per year was relatively low. This was also the time of the “green revolution” [29], characterized by the indiscriminate use of chemically synthesized fertilizers without considering their environmental and health impacts.

Between 1983 and 1990, there was a period of low productivity in research, but from the 1990s onward, there was a gradual rise in scientific output. Since 2006, there has been a consistent and steady increase in published scientific articles, reaching its peak in 2022. This remarkable surge in research activity can be attributed to the ongoing conflict between Ukraine and Russia, as these countries play a significant role as phosphorus fertilizer producers and promoters worldwide. The war has disrupted the supply of these fertilizers, leading to scarcity and, consequently, a growing interest in finding alternative approaches to address crop deficiencies resulting from reduced phosphorus fertilizer availability [30].

In a longitudinal analysis spanning four decades (1982–2023), 1168 scholarly articles were disseminated. The annual publication count reached a nadir in 1983, 1984, 1986, and 1987, with no articles published. Conversely, the zenith was attained in the antecedent year, 2022, marking the highest annual output.

The growing interest and research in solubilizing microorganisms since the 2000s have been driven by their potential biotechnological applications, benefits to soil health, and minimal environmental impact [31]. The field has seen exponential publication growth since 2018, indicating its maturation and significant impact on agriculture and environmental domains. The surge in research is also linked to the critical role of phosphorus, an essential macronutrient in plant growth, and its scarcity [32]. Given the increasing demand for food and sustainable agriculture, solubilizing microorganisms are being explored as a viable, efficient, cost-effective, and eco-friendly biofertilizer solution to address phosphorus limitations in crop production [33].

#### 3.1.3. Relevant Sources in the Domain

This section presents a comprehensive scientific and technical analysis of the top ten essential sources focused on the biotechnological utilization of solubilizing microorganisms to enhance phosphorus availability. Figure 4 displays valuable information regarding the number of articles published per source, their corresponding H-index, and CiteScore.

The H-index is a measure used to assess a journal’s citation productivity and impact. It considers the number of articles that have received at least the same number of citations [34]. On the other hand, CiteScore is a metric that represents the average number of citations per article derived from the Scopus database. The CiteScore is divided into quartiles, with Q1 indicating the highest level of influence and Q4 indicating the lowest [35].

The bibliometric analysis reveals that the journal *Science of the Total Environment* has the highest number of articles published in the field, with 43 articles accounting for 19% of the top 10 sources. Additionally, this journal boasts an impressive H-index of 317 and is placed in the Q1 quartile, signifying its position as the most influential and reputable journal in this research domain. Similarly, other high-impact journals, such as the *Journal of Environmental Quality*, *Frontiers in Microbiology*, *Frontiers in Plant Science*, the *Journal of Environmental Management*, and *Plant and Soil* also exhibit noteworthy H-index values above 180. These journals are placed in the Q1 quartile based on their CiteScore, further solidifying their impact and reputation within the field. In contrast, the *Journal of Soil Science and Plant Nutrition* possesses a comparatively lower H-index of 53 and is ranked in the Q1 quartile, indicating its relatively lower influence and reputation in this domain.

The ten journals contribute 223 articles, representing approximately 20% of the overall publications within this specific research domain. This finding underscores the significance of scientific production in high-impact sources and highlights the importance of research conducted in the exploration and biotechnological application of phosphorus-solubilizing microorganisms. This area emerges as novel and of great relevance within the scientific community.

##### Metrics of the Leading Journals in the Domain

In the analysis of the critical metrics of the most significant sources in the domain (Figure 5), it is observed that *Science of the Total Environment* and the *Journal of Environmental Quality* have considerable influence, as they have high H-indexes and occupy the top positions in terms of total citations and published articles. Similarly, *Frontiers in Microbiology* and *Frontiers in Plant Science* stand out as prominent sources due to their substantial m-indexes, which indicate the rapid attainment of high H-indexes soon after their initial publications. Despite being the newest in the table, *Microorganisms* is the most promising source in the domain, as it has the highest m-index and ranks high on a g-index, despite being the newest source in the table. Likewise, *Communications in Soil Science and Plant Analysis* and the *Journal of Plant Nutrition* are considered the least influential and impactful sources within this domain, as evidenced by their lowest H, g, and m indices, total citations, and published articles.

Furthermore, it is imperative to acknowledge the inverse relationship between the H-index and the quantity of published articles. Entities with elevated H-indexes typically possess fewer articles, signifying a prioritization of quality over quantity. Additionally, a correlation exists between the total number of citations and the year of initial publication, suggesting that sources accrue citations over time, thereby establishing their significance within the respective field as they mature.

#### 3.1.4. Relevant Universities in the Domain

This study reveals the publication trends in using solubilizing bacteria to enhance plant phosphorus assimilation (Figure 6). Zhejiang University leads with 47 publications, while Universiti Putra Malaysia has the fewest at 22. The leading universities are predominantly in China, Pakistan, India, Australia, and Malaysia.

In terms of country distribution, China has the highest number of universities (four) engaged in this research, followed by India (three), Pakistan, Australia, and Malaysia (one each). China also leads in the total number of publications (114), followed by India (80), Pakistan (37), Australia (27), and Malaysia (22).

The data suggest that using solubilizing bacteria as a biotechnological strategy to improve phosphorus assimilation in plants is a growing research trend, especially in Asian countries. This could be due to the limited availability of chemically synthesized phosphorus fertilizers, prompting the search for alternative solutions [36]. However, other factors, such as funding, infrastructure, policies, and each country’s specific context, may also influence research productivity and its correlation with national agricultural development and production models [37].

Turning to the scientific production of the leading universities in the domain (Figure 7), a noteworthy trend has been observed since 2009. There has been a steady increase in scientific production from these universities, indicating the emergence of a novel topic that initially received limited attention from the research community. Zhejiang University stands out with the highest research output among all relevant universities. Its research output has consistently remained high compared to others since 2010, showcasing its intense focus and significant contributions to the field. Conversely, the Faisalabad Agriculture University exhibits relatively lower research output compared to other universities, but there has been a gradual increase since 2019, suggesting a growing emphasis on this aspect.

Along this line, Zhejiang University is the primary contributor to research output on solubilizing bacteria as a biotechnological approach to improving plant phosphorus assimilability. Adelaide University and the Faisalabad University of Agriculture demonstrate different levels of involvement and increasing research output in this field. On the other hand, the Indian Council of Agricultural Research and Eternal University have limited or no research output in this area.

#### 3.1.5. Scientific Production of the Countries in the Domain

The global output of research on PSB, with the leading contributors being India, China, Brazil, Pakistan, and the United States, accounts for 52.6% of the total publications. These countries’ significant involvement is attributed to their large populations, extensive agricultural activities, and phosphorus-rich soils. The study of solubilizing bacteria is particularly relevant in these regions due to their extensive agricultural sectors and high demand for phosphorus fertilizers. As a result, these nations show a strong inclination towards adopting biotechnological alternatives to enhance plant growth and nutrient cycling [38].

The research productivity of various countries (Figure 8), including the United States, Australia, Japan, Morocco, Germany, and Egypt, is also highlighted. These countries, despite their diverse geographical, climatic, and economic conditions, demonstrate significant engagement in studying phosphorus-solubilizing bacteria, indicating their adaptability and versatility as practical biotechnological tools [39].

A comparative analysis reveals that Asia has the highest scientific yield in solubilizing bacteria research, reflecting the active participation of Asian nations in this domain. However, the contribution from American nations, such as Brazil, the United States, Mexico, Canada, and Colombia, is comparatively lower when juxtaposed with Asia, Europe, and other regions.

This paper also notes a substantial disparity between the most and least productive countries, with some regions needing more publications in this field. This discrepancy may be due to inadequate resources, infrastructural limitations, insufficient expertise, or lack of awareness of the potential benefits of solubilizing bacteria. The agricultural production patterns in these regions do not prioritize or necessitate extensive research on solubilizing bacteria.

##### Collaborations in the Domain Segmented by Country

In terms of the distribution of research publications on using solubilizing bacteria to enhance plant phosphorus assimilation [40], Figure 9 shows that these publications are from single countries, with cross-country collaborations accounting for only 15.3% of total publications. This suggests a low level of international collaboration in this research domain. Countries like India, China, the United States, Brazil, and Pakistan have many publications authored solely by their nationals, ranging from 75% to 92%. However, Japan and Spain, among the top 10 most productive countries, have at least 30% of their publications resulting from international collaborations.

Figure 9 also shows that there is potential for improving international collaboration in this field, given the value of the use of solubilizing bacteria as a biotechnological solution, especially in regions characterized by low soil fertility and significant phosphorus deficiency, which have a large niche for the development of this type of research. Therefore, multinational publications can facilitate the exchange of knowledge, resources, infrastructure, and expertise, which vitally contribute to improving the quality and impact of research [41]. It should be noted that the limited level of international collaboration observed in this domain can be attributed to several factors, such as language barriers, lack of funding opportunities, divergent research priorities and policies, and restricted access to data and resources.

##### Cross-Country Collaboration in the Domain

This section quantitatively examines the frequency of international collaborations among researchers utilizing solubilizing bacteria as a biotechnological approach to enhance plant phosphorus assimilation. The significance of this lies in the ability of these microorganisms to catalyze the transformation of insoluble phosphorus compounds into soluble forms, thereby facilitating their absorption by plants. Figure 10 provides a visual representation of the foremost collaborations, where the red lines indicate the relationship between countries, showing that the most recurrent collaborations are between China and Pakistan (14 instances), China and the United States (9 instances), China and Germany (6 instances), and India and Saudi Arabia (6 instances). Furthermore, China emerges as the most prolific collaborator globally, with 34 instances of multi-country collaboration. India is a close second, with 31 collaborations involving countries like Saudi Arabia and China. The United States is ranked third, with 27 collaborations spanning various countries globally, followed by Pakistan in the fourth position, with 21 collaborations.

The geographic distribution of research collaborations focuses on Asia, which has extensive collaborations with Europe and North America. In contrast, Africa, South America, and Oceania have fewer collaborations, with South America and Asia’s collaborations being particularly infrequent due to geographical, cultural, and language barriers [42].

Specific collaborations are highlighted, such as the frequent collaborations between China and Pakistan due to geographical proximity, historical ties, political alignment, and shared agricultural interests [43,44]. Similarly, collaborations between Brazil and Germany are significant in biotechnology and environmental sciences, while collaborations between China and Germany reflect their solid economic relationship and mutual interest in innovation and sustainability [45].

In Africa, intra-regional cooperation needs improvement due to limited resources for conducting and publishing research and inadequate access to international research networks [46].

Finally, there is the potential for increasing research synergies and inter-regional cooperation, particularly between regions at different development stages. This collaboration fosters the reciprocal transfer of knowledge and technology, specifically using solubilizing bacteria to enhance phosphorus bioavailability and promote plant growth.

#### 3.1.6. Domain Funding Sources

This section presents a comprehensive scientific analysis of the primary financial entities that enable the implementation of solubilizing bacteria as a biotechnological approach to enhance plant phosphorus uptake, as illustrated in Figure 11. The leading ten entities include four Brazilian institutions, two Chinese institutions, three Indian institutions, and one Canadian institution. The National Natural Science Foundation of China is the most significant contributor, supporting 59 academic papers. It is closely followed by Brazil’s Conselho Nacional de Desenvolvimento Científico e Tecnológico, which has sponsored 29 papers, and Brazil’s Coordenação de Aperfeiçoamento de Pessoal de Nível Superior, endorsing 24 papers. These findings highlight the significance of employing solubilizing bacteria to amplify phosphorus uptake across different countries, especially those with the highest publications in this domain. As a result, the research efforts of these nations are primarily focused on expanding knowledge in this field. Moreover, the funding sources include national bodies and regional foundations, emphasizing the broad acknowledgment of the research’s significance and potential influence within each respective country at national and regional scales.

In this regard, greater collaboration and communication between the different funding sources, sponsoring countries, and countries with fewer resources could be encouraged to improve the domain. This facilitates the exchange of knowledge, experiences, and resources on this topic and the identification of common challenges and opportunities for innovation. In addition, greater attention and support should be given to underrepresented countries or regions with the potential or need to use solubilizing bacteria as a biotechnological strategy to improve phosphorus assimilation in plants, such as Colombia or other Latin American countries.

It is also essential to indicate that more research should be conducted to explore the mechanisms, diversity, efficiency, and application of solubilizing bacteria in different soil types, crops, climates, and management practices. In addition, research is warranted to investigate this strategy’s environmental and socioeconomic impacts. These research efforts would contribute to a deeper understanding of the subject and pave the way for advances in this field.

#### 3.1.7. Conceptual Structure of the Domain

A specific research field’s theoretical framework is constructed to facilitate the understanding and investigate the interrelatedness of diverse concepts within that domain [47]. In this context, the emphasis is on examining solubilizing bacteria and their contribution to the augmentation of phosphorus uptake by plants via biotechnological methodologies.

The initial step involves a quantitative assessment of the domain’s predominant terminology. This involves thoroughly examining the most frequently utilized keywords in scholarly articles and those indexed in scientific databases. This analysis is crucial, as it identifies any discrepancies between the terminologies proposed by the authors and those algorithmically suggested by the databases.

A keyword frequency analysis of the scientific literature (Figure 12) related to using solubilizing bacteria to enhance plant phosphorus assimilation was conducted. The most frequently used keywords are “phosphorus” and “biofertilizer”, indicating a strong interest in improving phosphorus availability using microbial inoculants. Other prevalent keywords include “phosphate solubilization”, referring to the process of converting insoluble phosphorus into plant-assimilable forms, and “PGPR” (plant-growth-promoting rhizobacteria), a type of root zone that enhances plant growth and yield. The terms “plant growth promotion” and “sustainable agriculture” frequently appear, underscoring the literature’s objectives of enhancing plant productivity and quality through biotechnological approaches [48].

Regarding the keywords indexed by scientific databases, “phosphorus” remains the most prevalent term, aligning with the authors’ keywords and indicating its significant importance as per the databases’ algorithms [49]. A new keyword emerging from the database analysis is “soil”, which appears 267 times [38]. This previously unmentioned keyword suggests that soil is the primary medium for phosphorus solubilization and plant uptake. The third and fourth most frequently used keywords are “fertilizers” and “phosphate”, occurring 231 times. Together, these results imply a considerable demand in the literature to explore alternative and sustainable methods for improving the processes of phosphorus availability for plant or crop uptake.

Considering the above, the domain of study is well-defined, with a consistent focus on investigating the role of bacteria in facilitating phosphorus uptake. The central theme within this domain is the influence of phosphorus on plant growth, while other factors remain subjects for further examination.

##### Clustering According to the Thematic Association of the Domain

This section examines the thematic distribution of the domain, which refers to a visual representation showing the occurrence of specific words within the domain in the metadata of the analyzed documents. Topic maps classify these occurrences into four categories: driving, niche, emerging, and core topics. This classification is based on the concentration and importance of the topics [50]. Figure 13 illustrates the distribution of themes within the domain using a clustering technique that captures their associations.

In considering the primary themes of the domain, it is essential to highlight phosphorus as a central element of agriculture. Phosphorus is crucial in promoting plant growth, consistent with its essential nature for plant development [51]. However, the availability of phosphorus is limited due to its reactivity with other elements. To address this limitation, using solubilizing microorganisms can improve plant access to phosphorus, thus directly improving the agricultural process [52].

A significant and relevant relationship observed in the domain is the struvite precipitation process, which is closely related to the fundamental issues of this field; among the most crucial precipitation techniques are thermodynamic and kinetic modeling [53] and chemical precipitation [54]. Struvite, a mineral widely discussed in the scientific literature, is crucial for recovering nitrogen and phosphorus from anaerobic digestate [55]. Therefore, struvite precipitation offers an exciting method to recover phosphorus sustainably from sources like manure from various animals [56].

Furthermore, a correlation has been observed between work focusing on eutrophication, fertilization, and phosphorus availability. This has to do with the process of phosphorus extraction, and its subsequent transport in fertilizers, crops, and other products can lead to the accumulation of this mineral in certain soils around the globe. This accumulation generates elevated levels of phosphorus, which increases the likelihood of phosphorus runoff into aquatic ecosystems through erosion in lakes, streams, and rivers [57]. Excessive enrichment of nutrients and minerals, such as phosphorus, can cause problems, such as uncontrolled algal growth, thus negatively affecting aquatic tributaries and their ecosystems [58]. Therefore, managing these processes by using inorganic nutrients is essential in this field.

Another notable relationship among the various papers concerns PGPR plants. These bacteria are essential in sustainable agriculture, as they enhance plant growth through biological nitrogen fixation, phytohormone production, and phosphate solubilization [59]. Therefore, PGPRs constitute an attractive biofertilizer option that can contribute to the advancement of more sustainable agricultural practices.

The motor themes are well-developed and essential for structuring a research field [60]. Figure 13 indicates that the presence of motor themes is limited, indicating the domain’s novelty. More importantly, it highlights the need to establish solid conceptual foundations in this domain, which are consistently employed in various studies to avoid redundancy and thus incentivize advancement.

Niche topics refer to very specialized aspects of research in the field of phosphorus-solubilizing bacteria as a biotechnological alternative to improve plant assimilation [61]. One such topic focuses on the utilization of phosphorus to improve the overall agronomic efficiency of crops, which can be influenced by several factors, such as the solubility of phosphorus in water and citrate, the chemical composition of water-soluble solid phosphorus fertilizers, the various forms of liquid fertilizers, and the chemical reactions that occur when phosphorus fertilizers are applied to the soil [62].

Another niche topic within this field revolves around superphosphates, which offer a significant advantage as phosphorus-rich fertilizers. These fertilizers are used to replenish soil nutrient levels and promote plant growth. They also facilitate rapid root formation and growth, allowing plants to resist water shortages, especially in areas where availability is limited [63]. Another interesting relationship is between gluconic acid, in which PSB can convert insoluble phosphorus to soluble phosphorus, thus improving plants’ uptake and use of soil phosphorus. Of all organic acids, gluconic acid is the most frequent solubilizing agent for mineral phosphate.

As discussed in several articles, a vital connection observed in this field concerns the interaction between arsenic and groundwater. Excessive use of conventional phosphorus fertilizers, known for their ability to increase agricultural productivity, can lead to water pollution through eutrophication [17]. This phenomenon can lead to the accumulation of harmful substances in the soil, such as selenium and arsenic, thus affecting their concentration levels. There is another association between gluconic acid and PSB. These bacteria can convert insoluble phosphorus into a soluble form, thus facilitating plants’ uptake and utilization of phosphorus from the soil. Among all organic acids, gluconic acid is the most common mineral phosphate-solubilizing agent [64]. Another niche topic in the domain is the utilization of manure as a source of organic phosphorus. This makes possible the use of phosphorus-solubilizing bacteria, which are microorganisms that can convert insoluble phosphorus into a soluble form that plants can absorb. Although some advances are documented in the scientific literature, these have yet to be widely reported in this field of study [38,64].

##### Thematic Evolution of the Domain

Thematic evolution analysis is a systematic methodology utilized to identify, quantify, and visually represent the progression of a specific area of research. It achieves this through the implementation of clustering techniques and co-word network analysis. Cobo initially proposed this analytical approach [65]. A comprehensive examination of the scientific literature was conducted for the analysis, covering distinct periods: 1982 to 2012, 2013 to 2015, 2016 to 2020, and 2021 to 2023 (Figure 14). The division of these time frames was purposeful, aiming to reveal emerging trends within the field of study and to gather insights into future expectations. The core methodology employed to investigate thematic evolution involves the construction of concurrence networks. These networks categorize and interconnect frequently used words within scientific articles.

Substantial advancements were made in establishing the foundational principles of this research domain between 1982 and 2012. Several key terms were identified as forming the bedrock of this area, including nutrients, natural phosphate, plant growth promotion, soil, phosphorus, phosphate, and apatite [66]. These terms are of fundamental significance concerning phosphorus assimilation and closely aligned with the initial stages of research exploration.

During this early period, researchers likely concentrated on exploring the potential of specific bacteria to enhance the solubility of insoluble phosphorus compounds prevalent in soil, such as rock phosphate and apatite. Apatite, a phosphate mineral and the primary form of natural phosphate, is vital in determining phosphorus availability to plants, thereby promoting their growth and productivity [67]. Additionally, investigations into the interactions between microorganisms, soil properties, nutrient dynamics, and plant responses were likely undertaken about these bacteria.

Overall, the knowledge base established during this phase forms the groundwork for subsequent topics that will emerge as integral components of this research domain.

From 2013 to 2015, several terms, such as plant growth promotion, agriculture, biofertilizers, plant-growth-promoting bacteria, fertilizers, phosphorus, struvite, microbial biomass, and nutrients, constantly appeared in the scientific literature. Research during this period focused on investigating the potential of plant-growth-promoting bacteria, specifically in the form of biofertilizers. This term describes organic fertilizers derived from organisms that aim to improve nutrient availability and promote plant growth.

The mention of struvite indicates the scientific community’s interest in understanding the role of soil microbial biomass and phosphorus sources, such as struvite, in improving phosphorus assimilation [66]. The recurrent use of these terms in different articles suggests that the scientific community is actively studying how plant-growth-promoting bacteria improve phosphorus assimilation in plants. The goal is to develop efficient and sustainable agricultural practices while addressing the challenges associated with plant growth and agroecosystem sustainability.

When analyzing the relationship between these elements, it becomes evident that the term “plant growth promotion” has been constantly emphasized over the years in different research worldwide. In addition, it should be noted that minerals, such as apatite in the previous period and the use of struvite, speak to the concern to find new sources of soluble phosphorus to make up for the deficiencies of this element in the soil to avoid a loss of productivity in agricultural systems [68].

According to the analysis of the scientific literature published between 2016 and 2020, researchers have undertaken research on solubilizing bacteria to improve phosphorus assimilation in plants to understand which are the most efficient biochemical mechanisms in the transformation of insoluble phosphorus compounds into soluble forms, thus increasing its availability in the soil and, ultimately, avoiding a deficit of this element that allows for the correct growth of plants or crops [69]. The scientific community has explored various mechanisms related to phosphate utilization and investigated the potential advantages associated with the use of rhizobacteria and other types of bacteria that promote plant growth. An exciting development in this field is the emergence of nitrogen as a crucial new research element, reflecting the evaluation of hydrogen to understand the overall nutrient balance and its impact on plant growth. In addition, researchers have continued to emphasize the importance of minerals, such as struvite, as a promising source of phosphorus while expressing growing concern about the potential adverse effects of eutrophication. This concern has led to research into methods to promote efficient phosphorus uptake in plants.

The scientific literature between 2016 and 2020 suggests that using solubilizing bacteria as a biotechnological tool can improve plant phosphorus assimilability. These bacteria can solubilize phosphates by producing organic acids, such as citric and gluconic acids. In addition, they may have other PGPR properties, such as the production of Indole-3-Acetic Acid (IAA), a phytohormone that stimulates root growth. The composition of the bacterial community and the interactions between bacteria, fungi (such as *Aspergillus niger*), and the plant microbiome are crucial factors influencing the efficacy of solubilizing bacteria in sustainable agriculture [70]. Solubilizing bacteria can interact with other microorganisms and form biofilms in the soil, affecting the availability and cycling of other nutrients, such as nitrogen, by reducing dependence on chemical fertilizers and improving nutrient use efficiency. Applying solubilizing bacteria as biofertilizers holds promise for improving the productivity and sustainability of agroecosystems.

In detail, it can be found that one mechanism used by solubilizing bacteria to solubilize phosphorus is the secretion of organic acids, such as gluconic acid, citric acid, and oxalic acid. These organic acids can lower the soil pH and dissolve insoluble phosphates, such as hydroxyapatite and struvite, into soluble forms [32]. Hydroxyapatite and struvite are two common mineral forms of phosphorus in soils and wastewater. They are also potential sources of phosphorus recovery and recycling as fertilizers.

### 3.2. Execution of the Literature Review

The following is a solution to the above research questions to complement the domain analysis on using solubilizing bacteria as a biotechnological element to improve plant phosphorus assimilation.

#### 3.2.1. What Is the Importance of Phosphorus in Plant Physiology or Nutrition?

Phosphorus is a crucial macronutrient necessary for plant growth and function. It plays a fundamental role in several biological processes, such as the development of roots, grains, and flowers. Phosphorus is also involved in forming cell membrane phospholipids and genetic material. In addition, it is a vital component in photosynthesis, glycolysis, and fatty acid synthesis [71].

Despite its importance, the low availability of phosphorus in the earth’s crust limits agricultural production due to its low mobility in the soil, preventing the plant from absorbing it from the soil solution and having the capacity to grow without deficiencies of this element. For example, phosphorus deficiency can affect the biochemical reactions of the cell’s energy metabolism as it is an integral part of the molecules that accumulate energy as ATP. It is a result of photosynthesis and used in plant respiration. Therefore, generating new cells to produce roots at the beginning of vegetative cycles is vital. Compounds analogous to ATP, such as Uracil triphosphate, Guanosine triphosphate, and Cytosine triphosphate, are required to synthesize sugars, phospholipids, and ribonucleic acids [72]. Phosphorus is found in the soil pool in different forms, including oxygen-bound phosphates in rocks, which are diluted during weathering, allowing plant roots to absorb more quickly in organic and inorganic forms [73]. New alternatives are currently being sought to improve phosphorus uptake and take advantage of phosphorus sources present in the soil to reduce dependence on exogenous fertilization in agricultural production [33].

The phosphorus cycle in the soil is a complex and dynamic system that involves the accumulation of this element in the microbial biomass of phosphorus solubilizers, which include four general forms of phosphorus: available inorganic, organic, adsorbed, and as a primary mineral [6]. These different forms of phosphorus are recycled through water by the earth’s crust and living organisms in a sedimentary process characterized by the slow migration of phosphorus from deposits on land to marine sediments and its return to the soil and ocean. P is a fundamental component in the energy transduction processes, nucleic acid synthesis, and the creation of essential biomolecules. It is integral to the metabolic and physiological functions of plants. The presence of phosphorus promotes the development of a robust root system, facilitates appropriate flowering, and ensures efficient seed production. These factors are critical in agricultural practices to guarantee optimal crop yields. Phosphorus aids in accelerating maturation and enhancing stress resistance, attributes that improve the efficiency and sustainability of crop cultivation. Therefore, the effective management of phosphorus in agricultural practices is vital for maintaining plant health and optimizing the productivity and sustainability of our agricultural systems [33].

#### 3.2.2. What Are the Sources of Phosphorus Available in the World?

In the soil, plants can access two main types of phosphorus: inorganic phosphates and organic phosphates. However, plants are unable to absorb insoluble inorganic phosphorus compounds. The pH level of the soil influences the availability of inorganic phosphorus. Some insoluble inorganic phosphorus compounds include apatite, oxyapatite, struvite, and hydroxyapatite. Under favorable conditions, these compounds can dissolve and become accessible for plants to utilize [74]. Phosphate anions (PO_4_^3−^) are highly reactive and can be immobile through precipitation when combined with metal ions in the soil. In acidic soils, a substantial amount of phosphorus in fertilizer reacts with cations like Al^3+^ or Fe^3+^ through a precipitation reaction. This can result in the formation of orthophosphate ions, which plants can take up. Additionally, certain bacteria in the soil play a vital role in the process of solubilizing phosphorus, converting it into forms that plants can assimilate [33].

Phosphorus concentrations in soil solutions typically range from 0.1 to 1.1 mg·L^−1^. Among these concentrations, more than half may exist as soluble organic components released by dead cells or as colloidal organic components. Phosphate fertilizers are a type of simple fertilizer obtained from phosphate rock, a set of natural minerals containing a high concentration of P compounds. Phosphate rock is the primary raw material for producing this type of fertilizer, which does not generate environmental pollution like chemically synthesized fertilizers [75,76]. The P bioavailability in the pedosphere is contingent upon specific determinants, such as the soil’s pH value and particular bacterial taxa that facilitate the solubilization process, thereby rendering P accessible for plant absorption. Although fertilizers supplement the soil with phosphorus, the assimilation of this macronutrient may be constrained by the soil’s physicochemical properties. Conversely, phosphate fertilizers derived from phosphate rock represent a sustainable alternative due to their negligible contribution to environmental contamination [77].

##### Phosphate Rock

Phosphate rock deposits are composed of arenites with a diphosphorus pentoxide (P2O5) content ranging from 20% to 30% [76]. Soil fertility is limited by certain properties, such as pH; acid soils with high levels of Fe^+3^ and Al^+3^ cations have low plant-available phosphorus content and tend to fix phosphorus applied as phosphorus fertilizer. Consequently, phosphorus applied as phosphorus fertilizer is fixed in these soils, which reduces the effectiveness of water-soluble phosphate fertilizers, such as triple superphosphate and diammonium phosphate. In such cases, unprocessed phosphate rock presents an attractive alternative. RF contains calcium carbonates, which help achieve near-neutral soil pH levels. This, in turn, improves the availability of soluble phosphorus in acid soils, such as andosols and oxisols [78]. Conversely, limited solubility poses a significant constraint on phosphorus sources derived from phosphorite rocks, as corroborated by numerous studies conducted across various Latin American nations [79]. These investigations have demonstrated that the outcomes are not consistently favorable when phosphorite rocks are directly applied to the soil, either in their intact or pulverized state, particularly in short-duration crops and in soils with alkaline pH, where phosphorus tends to be sequestered by soil colloids [80]. However, finely pulverized phosphorite rock may gradually convert its phosphorus to a soluble form in the soil. This process is influenced by soil acidity and phosphorus-solubilizing microorganisms, which vary among different soil types. This is essential to augmenting phosphorus availability and ensuring adequate crop nutrition, both short-cycle and perennial [81].

##### Phosphorus Availability in Plants

Phosphorus is an essential element for the growth and development of plants and microorganisms, as it plays a crucial role in the mixture of energy accumulation and release processes during cell metabolism. However, it is essential to consider that soluble phosphorus is limited in agricultural production and natural ecosystems [78,82]. Moreover, its availability depends on the soil type, as approximately 50 to 60% is found in an organic fraction, while the rest is in inorganic form [83]. Plants take up soluble inorganic phosphorus, but when introduced into the soil in percentages above 90%, it is rapidly converted into forms unavailable to plants [84]. Therefore, much of the applied soluble phosphate fertilizers are not used by plants but are stored in the soil or become sources of pollution of water bodies [85].

It is important to note that phosphorus is found in the soil in non-nutritious forms for plants, which must assimilate the element from the soil even though it is found in low concentrations [84]. Typically, phosphorus levels in soil vary between 5 and 30 mg kg^−1^, representing low nutrient ratios. In alkaline soils, phosphorus reacts with cations, such as Ca^+2^, Fe^+3^, and Al^+3^, which causes its precipitation or fixation, decreasing its availability to meet the needs of plants or crops [86]. On the other hand, it should be mentioned that phosphorus fertilization in agricultural systems significantly increases production costs [87]. For this reason, several studies focus on the use of the reserves that the soil has in non-assimilable forms to transform them into usable forms; one of the alternatives is the use of microorganisms called phosphorus solubilizers. Their mechanisms of action between plant microorganisms have to be an effective relationship so that the availability of phosphorus has positive effects on crops [86,88].

#### 3.2.3. How Is Phosphorus Availability Improved Using Phosphorus-Solubilizing Microorganisms?

The escalating requirement for sustenance, propelled by demographic augmentation and escalating affluence, necessitates amplified crop yields. Regrettably, anthropogenic interference has precipitously diminished the fecundity of agrarian soils, particularly P, through overutilization, thereby jeopardizing production and augmenting the global risk of famine [89]. P is indispensable, yet its limited bioavailability poses a challenge in over 40% of global soils, exacerbating this issue through overexploitation due to the intensification of agricultural production systems. Without corrective measures, phosphorus reserves could be exhausted within a few decades. 

Phosphate fertilizers address P deficiencies in crops, but plants can only utilize between 5% and 25% of their content. Upon interaction with certain elements, such as Ca^2+^, Al^3+^, and Fe^3+^, and acidic pH, these compounds rapidly become inaccessible to plants, which is where PBS can intervene. These microorganisms enhance P bioavailability in the soil and promote plant growth, representing a sustainable and efficacious solution [90]. Despite the remarkable potential of PBS, it is crucial to comprehend its impact on plant nutrition. These bacteria can capture phosphorus retained in the soil and convert it into soluble forms [74]. They accomplish this by releasing various organic acids into the soil, facilitating phosphorus solubility and its availability to plants (as will be elaborated in subsequent sections).

Phosphorus-solubilizing microorganisms are capable of releasing phosphorus into soils through biological solubilization mechanisms. This process promotes plant growth and has been extensively documented in the scientific literature, with phosphate release mediated by phosphorus-solubilizing microorganisms being one of the most detailed mechanisms [86]. PSB microorganisms are concentrated in the rhizosphere and metabolically active in the zone near or inside of the root [91]. Typically, amounts ranging from 10^1^ to 10^10^ CFU/g soil can exceed 2000 kg ha^−1^ of microbial biomass. Bacteria are exceptionally diverse in this ecosystem, with a wide range of genera, such as bacilli, cocci, and spirochetes, among others [92]. PSB communities are ubiquitous, with variations in forms and populations in different soils, and depend on the physicochemical properties of phosphorus content and cultural activities of agricultural systems. In addition, it has been shown that herbaceous plants present more abundant PSB communities than perennial plants [93]. It has been reported that these communities increase their populations in agricultural systems, especially in soils cultivated with cereals [94].

Soluble organic phosphorus is mineralized by nonspecific extracellular phosphatases in PSB microorganisms [95]. These microorganisms can break phosphodiester or phosphoanhydride bonds present in organic matter. In addition, there are specific phosphatases, such as phytases, whose role is to release phosphorus from phytic acid, one of the primary forms of organic phosphorus reserves in the soil [96]. The plant recruits PSB microorganisms in the rhizosphere zone, and part of them enters the root tissue. This complex biological interaction allows plants to have sources of soluble phosphorus from organic matter or inorganic substances while providing them with organic substances (secondary metabolites) for their survival [97].

In P solubilization and mobilization, PSB microorganisms play a fundamental role as they are responsible for solubilizing about 40% of the total phosphorus in the soil. The taxonomic genera that stand out are *Bacillus*, *Pseudomonas*, *Enterobacterales*, *Burkholderia*, and *Rhizobium* [98,99]. Phosphorus solubilization and mobilization by microorganisms is an indispensable mechanism for promoting plant growth, as numerous studies have shown that plant responses to inoculation by these microorganisms contribute to nutrition and improve agricultural systems [76,100]. The groups of microorganisms that stand out for their ability to solubilize phosphorus are shown in Table 3.

Microorganisms present in the soil play an active role in mineralizing organic phosphate. Laboratory studies have shown that microorganisms of the PSB group utilize various sources of organic phosphorus to meet plant needs and can also accelerate the incorporation of this element into the soil solution for storage [114]. Microbial activity plays a crucial role in transforming organic matter in the soil. However, the amount of mineralized phosphorus and its relative contribution to plant nutrition in agricultural systems still need to be fully understood due to the need for more research in this field [115]. Some microorganisms of the PSB group utilize phosphorus-rich organic substances, such as phytates, for phosphorus solubilization and mineralization, making them assimilable by plants [116]. Phosphorus availability for plant nutrition can be improved by inoculating microorganisms producing phytase enzymes or through exogenous addition of phytases to roots, especially when phytate is present in the substrate [116,117]. However, a more significant response to inoculation has been observed in plants with large amounts of phytate.

The solubilization of inorganic phosphorus produces and releases to the medium siderophores, hydroxyl ions, and organic acids, such as acetic, gluconic, oxalic, succinic, and malic acids, among others. These acids of low molecular weight, in addition to lowering the pH of the medium, act as fundamental chelators for the solubilization of inorganic phosphorus, which forms complexes with Ca^+2^, Fe^+3^, and Al^+3^ cations. Several microbial groups produce enzymes, such as phosphatases, C-phosphorus lyases, and phytases, to mineralize organic phosphorus, releasing different forms of organic phosphorus into the soil solution through hydrolysis of phosphodiester bonds [118].

In general terms, phosphorus availability in the soil for plant nutrition is regulated by three main processes that affect its concentration in the soil solution: dissolution/precipitation, adsorption/desorption, and interactions between phosphorus in solution and soil solid surfaces and mineralization/immobilization, which are biologically mediated conversions between organic and inorganic forms of phosphorus [119].

Inorganic phosphorus is present in the soil solution as orthophosphate anions, coming from the mineralization of organic materials and the solubilization of mineral sources [118], whose concentration varies over time due to environmental conditions and the availability of organic matter and parent material that is rich in inorganic phosphorus. The concentration of orthophosphate anions in the soil solution ranges from 200 to 5000 mg phosphorus kg^−1^ [115,118].

Finally, mycorrhizal symbiosis is an ancient mutualistic relationship between fungi of the order Glomales (Zygomycetes) and the root systems of most plant species. This symbiotic relationship, which dates back some 398 million years, predates the symbiotic interactions of plants in the family Fabaceae [120]. It is estimated that about 90% of all terrestrial plant species participate in this symbiosis, which plays a crucial role in the functioning of global natural ecosystems [121]. In this symbiotic relationship, plants provide fungi with carbon and secondary metabolites derived from their photosynthates. In return, the fungi enhance the plants’ ability to absorb phosphates and other minerals from the soil and produce phytohormones that increase the root surface area and strengthen resistance against soil pathogens [122].

A significant challenge for mycorrhizologists is deciphering the signaling mechanisms that enable efficient colonization of host plants by these fungi. Several methodologies have been developed to study these interactions and cultivate mycorrhizal fungi under laboratory conditions. These include traditional soil-based systems, aeroponic and hydroponic methods, and the recent in vitro root organ culture technique [122]. It is known that about 150 species of mycorrhizal fungi can colonize approximately 225,000 plant species. This broad colonization capacity indicates that arbuscular mycorrhizal fungi are not limited to symbiotic relationships with a single plant family, unlike in the Fabaceae family [123]. The high adaptability of these fungi and their integration into many plant species support the use of mycorrhizal-fungi-based biotechnologies to improve agricultural practices and manage crops in diverse plant environments, highlighting the importance of symbiotic relationships to promote sustainable agricultural practices [120].

To mitigate low P availability, plants developed different strategies, such as adaptive biochemical and symbiotic mechanisms, to increase inorganic phosphorus acquisition and improve the efficiency of its internal utilization [115,124]. One of the most important is the biological association of the root with free-living soil microorganisms capable of solubilizing organic and inorganic phosphorus sources, respectively [125].

#### 3.2.4. What Enzymes or Mechanisms Do Phosphorus-Solubilizing Microorganisms Produce So That the Plant Can Absorb Them?

Several processes, primarily dissolution–precipitation, and sorption–desorption, influence the phosphorus concentration in the soil. The primary mechanism by which soil microorganisms solubilize phosphorus involves the release of complex compounds or mineral solvents. These include organic acid anions, siderophores, protons, hydroxyl ions, and carbon dioxide. Additionally, various extracellular enzymes are released, further facilitating the solubilization process. This intricate biochemical interaction plays a crucial role in the phosphorus cycle within the soil ecosystem [125,126].

##### Organic Phosphate Mineralization

Plants cannot directly take up high-molecular-weight organic forms of phosphorus, which are generally transformed into inorganic forms after mineralization. Organic transformation and mineralization are catalyzed by different soil enzymes, mainly phosphatases that hydrolyze organic phosphorus into its inorganic forms, such as HPO_4_ and H_2_PO_4_, so plant roots can absorb it through the soil solution. Phosphatases can hydrolyze both phosphate ester and anhydride bonds, including acid and alkaline phosphatases, diadenosine triphosphatases, exonucleases, phosphoprotein phosphatases, phosphodiesterases, 5′-Nucleotidase, phytases, and acid phosphomonoesterase. The mineralization of organic phosphorus is carried out by microorganisms in the soil and influenced by environmental factors and exceptionally moderate alkalinity. In soil, organic phosphorus is found in the form of inositol phosphate, phosphodiester, phospholipids, nucleic acids, pesticides, antibiotics, detergent additives, and other forms resistant to chemical hydrolysis by soluble forms [127]. Phosphatases are produced by many soils’ microbial species, including *Aspergillus*, *Bacillus*, *Mucor*, *Penicillium*, *Rhizopus*, *Pseudomonas*, and *mycorrhizal hyphae*. Acid phosphatases predominate in acid soils, while alkaline phosphatases predominate in neutral and alkaline soils. Plants and phosphatases release orthophosphate ions from organic P forms, making microbial phosphatases more efficient [126,128].

Regarding the mineralization of inorganic phosphorus, one of the most prominent mechanisms is the production of organic acids by microorganisms in the soil. These acids lower the pH of the soil rhizosphere or complex the cations responsible for the precipitation of phosphorus. These acids can compete with phosphorus for sorption sites in the soil or form soluble complexes with metal ions associated with insoluble phosphorus compounds, such as AlPO_4_, Ca_3_(PO_4_)_2_, and FePO_4_, to dissolve mineral phosphorus [33].

##### Genetic Mechanisms of Phosphorus Solubilization

Some studies suggest that organic matter may function as the main reservoir (30–80%) of immobilized phosphorus, mainly in forms like orthophosphoric acid, phytic, inositol phosphates, phospholipids, and nucleic acids, which are degraded by the activity of enzymes, such as acid phosphatase, phytase, and nucleases, allowing plants or crops to absorb this element [33].

##### Organic Phosphorus Solubilization by Phosphatases

Organic phosphorus solubilization in soil occurs through the action of the enzyme phosphatase, which degrades macromolecules to micromolecules and converts them into water-soluble sources of phosphorus. These sources are transported through the soil pore network and recognized by the root systems of plants, biota, and microbiota [129]. P can be released from organic compounds by the enzymatic action of three groups of enzymes. The first group, the nonspecific phosphatases or phosphomonoesterases, performs dephosphorylation processes through a biochemical reaction that extracts phosphorus from phosphodiester or phosphoanhydro bonds in organic matter. The second group of enzymes, phytases, releases phosphorus from phytic acid. The third group comprises the enzymes phosphatase and C-P lyase, which break the C–phosphorus bonds in organic phosphonates [33,115].

Phytase enzymes are a type of phosphatase produced by microorganisms, such as *Bacillus* sp., *Enterobacterium Burkholderia*, and *Azo spirillum*, among others [130]. They allow for the release of insoluble phosphorus from the hydrolysis of phytates present in the phytic acid (*inositol hexaphosphate*) available in the soil. Phytic acid also acts as a storage source of Mg^+2^ and K^+^, ions, and, to a lesser extent, Ca^+2^, Mn^+2^, Ba^+2^, and Fe^+2^.

The solubilization of organic phosphorus is a process catalyzed by enzymes. Among these enzymes, the most studied are phosphatases, which participate in the dephosphorylation of phosphodiester groups attached to organic matter, and phytases, which catalyze the hydrolysis of phytic acid, sequentially releasing up to six free orthophosphate groups. Phytase enzymes, or Myo-inositol hexaphosphate hydrolases, are of great scientific and commercial interest due to their wide natural distribution in plants, microorganisms, and some animal tissues [115].

The activity of phytase enzymes is higher in microorganisms, such as Gram-positive bacteria (*Bacillus*), Gram-negative bacteria (such as *Klebsiella* [131] and *Pseudomonas* sp. [132]), filamentous fungi (such as mycorrhizal fungi and *Aspergillus niger*), and other phylogenetic groups of bacteria [133]. Mainly, these enzymes are produced by Gram-negative bacteria in the intracellular zone of the periplasm, whereas in Gram-positive microorganisms, they are extracellular. Acid phytases with a pH between 2.5 and 5.5, belonging to the acid histidine phosphatases family, have also been identified. For soluble phosphorus to be produced by bacteria colonizing plant tissues, especially the root surface and interior, these microorganisms must become established in these tissues [134].

In Figure 15, derived from Israel da Silva’s study [135], the mechanisms of phosphate solubilization are illustrated, demonstrating the phosphorus cycle and its mobilization. The base of the arrows contains numbers and symbols, each corresponding to a specific process involved in phosphorus mobilization, as detailed in the figure’s heading. These processes include Non-Specific Acid Phosphatases (NSAPs, 4.1.1), phytases (4.1.2), phosphatases (4.1.3), carbon–phosphorus lyases (C-P Lyases, 4.1.4), organic acids (4.2.1), inorganic acids (4.2.2), enzymes or enzymolysis (4.2.3), siderophores (4.2.4), exopolysaccharides (4.2.5), proton release (4.2.6), hydrogen sulfide (H2S) production (4.2.7), and direct oxidation (4.2.8). Each number corresponds to the section where the respective mechanism is explained in detail.

#### 3.2.5. What Are the Colonization Processes in Rhizosphere Phosphorus-Solubilizing Microorganisms?

Plant–microorganism interactions are highly complex and dynamic. They start with the microbial colonization stage, which can occur on different plant surfaces, such as the rhizosphere and endophyte [136].

##### Rhizosphere Colonization

The rhizosphere is the soil area surrounding plant roots, and it is highly influenced by them. This environment is dynamic and has a great diversity of microorganisms [137]. Rhizospheric colonization begins with the migration of bacterial populations into the roots in response to the release of organic compounds from plants into the soil, known as photosynthates or exudates, which contain essential nutrients, such as amino acids, nucleotides, fatty acids, organics, phenolics, sugars, and vitamins, that are used by microorganisms as a source of nutrients for their growth [138]. The “rhizosphere effect” involves the capacity of plant exudates, which act as chemoattractants of beneficial microbial communities and thus increase microbial densities in concentrations in the order of 108 to 1012 UFC/g soil, while bacterial concentrations in soil regions that are far from the root system of plants present concentrations of less than 106 UFC/g soil [139].

Once rhizospheric microorganisms establish contact with the root surface, they can form microcolonies and, depending on the type of microorganism, remain on the surface or progress to endophytic colonization [139]. Bacteria have been reported to respond to plant exudates by expressing various genes, such as those associated with synthesizing exopolysaccharides for biofilm formation [140], which protect microbial communities from adverse environmental factors [141].

*Bacillus* subtilis is attracted to L-malic acid secreted by *Arabidopsis thaliana*. This compound activates biofilm formation in a process that depends on the same genes required for biofilm formation in vitro [141]. In the case of *Azospirillum brasilense*, it is a PSB-type soil bacterium that promotes the growth of grasses, such as wheat and corn, colonizes the rhizoplane, and forms biofilms that allow it to compete for space with indigenous soil microorganisms [142]. On the other hand, the formation of biofilms with *Paenibacillus polymyxa* in the roots of *Arabidopsis thaliana* and barley has been reported, demonstrating the colonization pattern by using a strain labeled with GF-phosphorus protein and microscopic techniques. There are several pieces of evidence suggesting the crucial role that *exopolysaccharide* production plays in biofilm formation in the early stages of plant–microorganism interaction and allows it to compete with other microorganisms that are colonizing the rhizosphere and rhizoplane zone of the root [143]. This interaction is essential in determining the effects of plant growth promotion; without it, growth-promoting soil bacteria cannot improve plant nutrition or stimulate growth by producing phytohormones. Therefore, on many occasions, biofertilizers do not have positive effects due to the lack of plant–microorganism interaction [144].

##### Endophytic Colonization

Endophytic colonization refers to the colonization of internal plant tissues by bacteria called endophytes. These microorganisms can be isolated from plant tissues with surface disinfection, do not cause visible damage to the plant, and can promote growth [145]. Conceptually, endophytic bacteria are defined as those bacteria that can be isolated from plant tissues with surface disinfection and do not cause visible damage to the plant. However, this definition does not include non-culturable endophytic bacteria because of experimental limitations [146]. Most endophytic microorganisms come from or are recruited from the rhizosphere environment community. Therefore, they must be efficient colonizers of the rhizosphere and rhizoplane [147]. Successful endophytic colonization implies compatibility between the host and the endophyte. Furthermore, it is suggested that endophytes are specialized members of the rhizoplane colonizing community and that the plant selects the bacteria most adapted to endophytic life from a large pool of rhizospheric bacteria. Although it is unknown whether endophytic bacteria need to colonize a specific tissue or organ to perform their functions adequately, it is postulated that stochastic, environmental events and bacterial-dependent factors influence their ability to colonize internal plant tissues.

Concerning spatial distribution within the plant, although a more significant presence of endophytic bacteria is generally observed in roots, their colonization has been reported in other organs, such as stems, seeds, leaves, fruits, tubers, reproductive organs, and nodules [148]. Endophytic bacteria from the rhizoplane to the root cortical tissue can enter through passive and active mechanisms and various routes of entry [146]. Passive entry can occur through natural cracks in the lateral root emergence zones (via the middle lamina of the epidermis), through the base of root hairs and in the root apex growth zone, or by those caused by other microorganisms [149]. Several studies have reported the colonization of endophytic bacteria in the root through root hairs. Other sites commonly used by endophytes to colonize plants are the stomata, lenticels, and radicles [150]. Various entry mechanisms exist for microorganisms to penetrate plant tissues, with one involving active processes that rely on lytic enzymes capable of breaking down the plant cell wall. Among these enzymes are pectinases and cellulases, which play a crucial role in creating fissures or cracks in the rhizodermis, thus providing a pathway for the microorganisms to enter. The enzymatic activity appears to be facilitated by passing through the endodermis, which enables penetration into the pericycle and xylem vessels [150,151]. Notably, the ability to degrade pectins found in the primary plant cell wall and lamina media is a shared characteristic among many plant-associated bacteria, including endophytic ones.

Endophytic bacteria employ active mechanisms to penetrate plant tissues, which involve the production of lytic enzymes that degrade the plant cell wall. These enzymes, such as pectinases and cellulases, favor the formation of fissures or cracks in the rhizodermis, through which the microorganisms enter. The enzymatic activity seems to be mediated by passage through the endodermis, which allows for penetration into the pericycle and xylem vessels [150,151]. The ability to degrade pectins of the primary plant cell wall and lamina media is a common feature of many plant-associated bacteria, including endophytic bacteria.

The primary distinction between pathogenic and endophytic bacteria resides in the type of lesions they induce in the plant tissues they inhabit. Additionally, the ability to secrete cellulolytic enzymes plays a crucial role in enabling the breakdown of the plant cell wall, thereby facilitating vertical dissemination. For instance, *Azoarcus* sp. strain BH72 does not rely on cellulases for its growth, as it cannot utilize Carboxymethyl cellulose as the sole source of carbon and energy. Consequently, its function may lie in promoting the invasion of tissues during root colonization.

Unlike phytopathogenic organisms, which cause aggressive damage to plants, it has been shown that *Azoarcus* sp. does not secrete its cellulolytic enzymes but that these are associated with its cell surface. Thus, it is postulated that they may mediate more localized and less damaging plant cell wall digestion. Hurek et al. [152] reported that *Azoarcus* sp. can enter the intercellular spaces of the grassroots and increase cell density, mobilizing between the central vascular tissues of plants until reaching the aerial tissues, such as stems and leaves, where it generates growth promotion and phosphate solubilization. These findings have been reported in several studies [93,115,132].

#### 3.2.6. What Are the Prospects for Phosphorus-Solubilizing Microorganisms as a Biotechnological Solution?

Integrating phosphorus-solubilizing microorganisms into sustainable agricultural practices begins with identifying and selecting strains that exhibit high phosphorus solubilization efficiency [153]. This requires a literature review to determine strains previously reported for their phosphorus-solubilizing ability. In addition, it is imperative to isolate and characterize new strains from diverse sources, such as soils, rhizospheres, endophytes, or phyllospheres of crops or other plant species [154]. This process involves the evaluation of phosphorus solubilization capacity under controlled laboratory conditions, which facilitates the selection of strains that demonstrate superior solubilization potential [135].

Following laboratory selection of phosphorus-solubilizing strains, culture conditions should be optimized to increase their proliferation and efficiency, including determining the optimal pH, temperature, and nutrient conditions that promote microbial growth and translating these conditions from laboratory to large-scale culture [155]. Secondary metabolites produced by solubilizing microorganisms, which play a role in phosphorus solubilization, should also be evaluated. Once optimal growth conditions for the microorganisms have been achieved, controlled trials with various agronomic interest crops and different soil types will be conducted [156]. These experiments should be designed to evaluate the appropriate dosage for each crop, evaluate soil persistence, and determine the solubilization efficiency in different soil types. Applying selected microorganisms will allow for the measurement of phosphorus availability in the soil and its impact on crop growth and yield, providing crucial data to fine-tune application methods prior to field trials [157].

A critical step towards the large-scale application of this biotechnology is the development of field trials to validate the biotechnology and evaluate the efficacy of phosphorus-solubilizing microorganisms under natural conditions. Experimental plots of each soil type, climatic condition, and dose are needed. Treatments with phosphorus-solubilizing microorganisms should be applied together with traditional agricultural practices to evaluate their impact on soil phosphorus availability, plant growth, and crop yield [158]. This step is vital to confirm the practical feasibility and the economic and environmental benefits of this biotechnology, as the high solubilization efficiency often observed in the laboratory may not translate into significant effects on crops under natural conditions [159].

After field confirmation of the efficacy of the solubilizing microorganisms, the next step is developing the commercial product. This process involves investigating different formulations (liquid, solid, encapsulated) that maximize the viability and efficacy of these microorganisms [160]. The formulations developed must also undergo stability and shelf-life testing to ensure their quality over time and to obtain the necessary regulatory approvals by national and international standards, which are essential for commercializing these products domestically and internationally [161].

The successful adoption of phosphorus-solubilizing microorganism biotechnology in agriculture may depend on effective training and technology transfer to small, medium, and industrial farmers [17]. To adopt this biotechnology, it is crucial to develop training programs and educational materials that elucidate the use and benefits of the product, both from an economic and environmental point of view. In this regard, workshops, seminars, and field demonstrations should be organized to familiarize farmers with these new technologies and establish collaborative networks with agricultural cooperatives and rural development organizations, which is also essential to facilitate adoption [161].

Finally, monitoring systems must be established to assess the long-term impact of the application of this biotechnology in agriculture. This includes monitoring the availability of phosphorus in the soil and crop yields and reducing the use of chemical fertilizers. These data will allow for the assessment of economic and environmental impact and facilitate the publication of reports and case studies highlighting successful experiences and lessons learned with the developed biotechnology product.

## 4. Threats to Validity

This systematic literature review on phosphorus-solubilizing microorganisms as a biotechnological alternative identified and addressed several potential threats to validity. These threats are categorized into four main types: internal validity, external validity, construct validity, and conclusion validity.

Selection bias

The investigators’ subjective judgments may have influenced the selection of primary studies, leading to the inclusion or exclusion of relevant studies. To mitigate this, rigorous and predefined inclusion and exclusion criteria were employed, and several investigators independently reviewed studies to reduce individual bias.

Language bias

The review focused primarily on English-language publications, which could overlook valuable research published in other languages. Although we acknowledge this limitation, we included studies with abstracts in English regardless of the language of the full text to mitigate this.

Temporal validity

The rapid advancement of biotechnology may make some older studies less relevant to current practices. To mitigate this, we analyzed the publication dates of included studies and gave more weight to recent research while still recognizing foundational work.

Researcher expectations

Researcher expectations could inadvertently influence the interpretation of results. To mitigate this, we employed multiple reviewers for data extraction and interpretation and used standardized forms to ensure consistency.

By recognizing and addressing these threats to validity, we aim to improve the robustness and reliability of this systematic review. We recognize that eliminating all threats is impossible, but by transparently reporting these potential limitations, we provide readers with the context necessary to interpret our findings critically. Future research should address these limitations to advance our understanding of phosphorus-solubilizing microorganisms as a biotechnological alternative.

## 5. Conclusions and Future Work

This study involved conducting a bibliometric analysis and literature review to examine the status of the domain of phosphorus-solubilizing microorganisms and their role in improving phosphorus availability for plant uptake. Different bibliographic databases were used to collect the different scientific papers, totaling 2322 research papers. Subsequently, the methodology proposed by Donthu and Jia’s guide was used to ensure a systematic and rigorous approach to the literature review. The bibliometric tools, such as R-studio and Bibliometrix, facilitated the visualization and analysis of the data. The results indicate a growing interest in research on these microorganisms since the 2000s, highlighting their importance in the context of sustainable agriculture and phosphorus scarcity. Notably, India leads the research output with 542 published papers, followed by China with 336, Brazil with 225, Pakistan with 217, and the United States with 158 publications. Together, these countries account for 52.6% of the total publications, indicating their significant contribution to the domain; this may be due to their extensive agricultural sectors and demand for phosphorus fertilizers. The analysis also showed relatively limited international collaboration, suggesting the possibility of further collaboration between countries.

The study also provided information on phosphorus solubilization mechanisms and the main microorganisms involved. Phosphorus is an essential macronutrient for plants, but it has reduced availability; this situation can be overcome by releasing organic acids, phosphatases, phytases, and other compounds that liberate phosphorus from insoluble compounds. Several genera of bacteria can solubilize phosphorus, especially *Bacillus*, *Pseudomonas*, *Burkholderia*, and *Rhizobium*. These microbes undergo complex *rhizospheric* and *endophytic* colonization processes to interact with plant roots and deliver phosphorus. However, to achieve phosphorus solubilization and ensure that it is efficiently provided to plant species, it is required that PSB microorganisms develop an effective interaction with plants to favor the growth of this type of microorganism. For this, the plants, and especially the root, must provide secondary metabolites. If this interaction fails to be effective, as it has been evaluated for several plant-growth-promoting inoculants, plants cannot obtain soluble phosphorus, decreasing their nutrition.

In addition, it was found that the use of alternative methodologies to find solutions to phosphorus deficits in soils is promising due to the high diversity of phosphorus-solubilizing microorganisms with the biotechnological potential to transform insoluble phosphorus, which is reserved in soils as soluble phosphorus, contributing to the improvement of plant growth and development. In this way, farmers can lower the production costs of their crops, in addition to having cleaner production, as they avoid the excessive use of chemical fertilizers, which contain heavy metals, strong acids, such as nitric acid, beneficial acids in the soil, contaminate aquatic ecosystems, increase their acidity, and ultimately interfere with plant growth, thus affecting environmental dynamics.

This research highlights phosphorus-solubilizing microorganisms’ potential as a sustainable alternative to conventional phosphorus fertilizers. However, further research is needed on these microbes’ diversity, efficacy, and applicability in different environments and crops. Further international collaboration and research on microbial mechanisms can pave the way for effectively integrating this technology into agricultural systems. The results of this study provide a valuable basis for guiding future research in this emerging field.

Finally, regarding future work, one could talk about exploring the diversity of efficient phosphorus-solubilizing microorganisms and their bioprospecting in diverse environments, elucidating the molecular pathways involved in the production of organic acid, phosphatase, and phytase, and the study of synergistic effects between solubilizing microorganisms and mycorrhizae.

## Figures and Tables

**Figure 1 microorganisms-12-01591-f001:**

The methodology used in the literature review.

**Figure 2 microorganisms-12-01591-f002:**
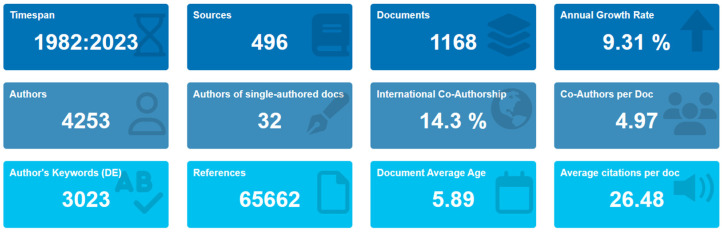
General information on the domain of solubilizing bacteria as a biotechnological tool.

**Figure 3 microorganisms-12-01591-f003:**
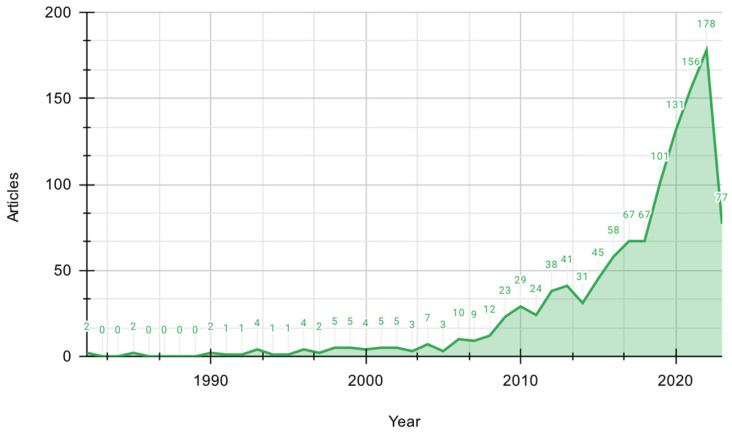
Scientific production in the field of solubilizing bacteria as a biotechnological tool.

**Figure 4 microorganisms-12-01591-f004:**
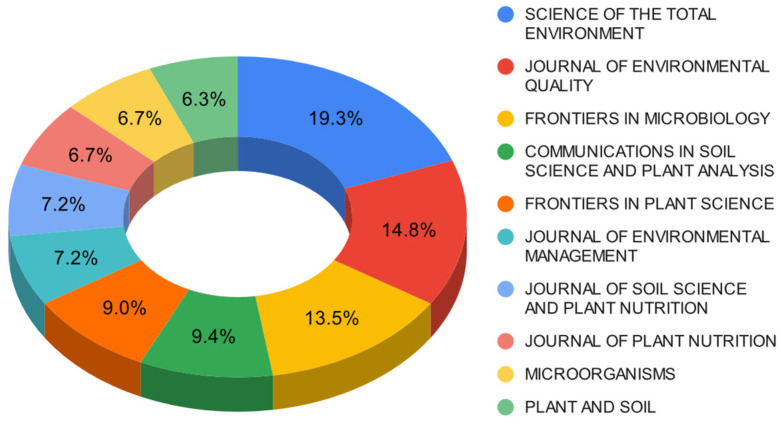
Primary sources in the domain.

**Figure 5 microorganisms-12-01591-f005:**
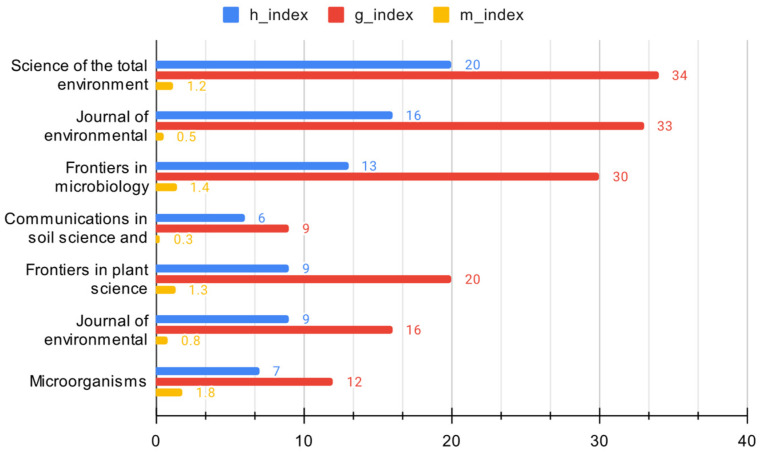
Primary sources’ productivity metrics in the domain.

**Figure 6 microorganisms-12-01591-f006:**
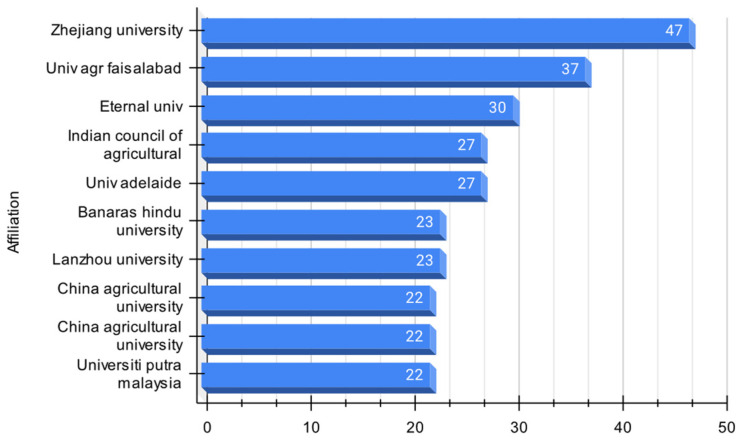
Central universities contributing the most to solubilizing bacteria as a biotechnology approach.

**Figure 7 microorganisms-12-01591-f007:**
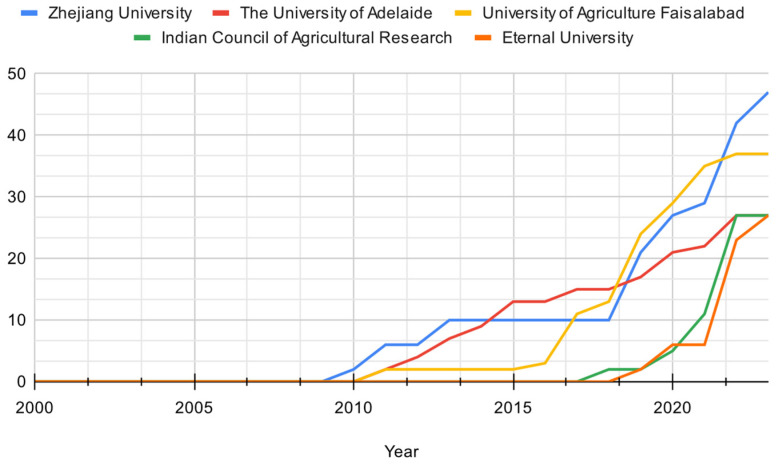
Chronological evolution of the research production of the relevant universities in the domain.

**Figure 8 microorganisms-12-01591-f008:**
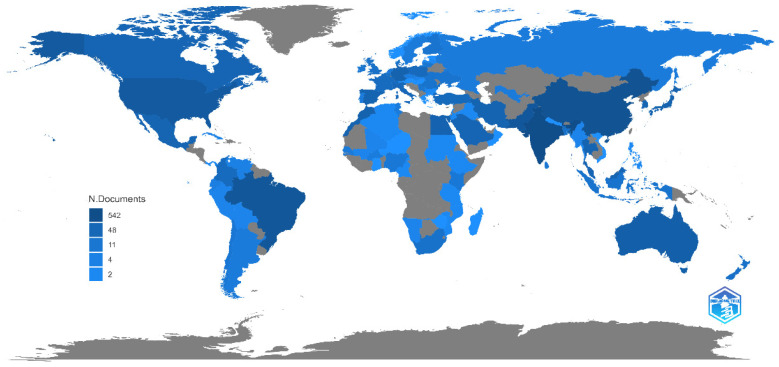
Scientific production is segmented by country in the domain.

**Figure 9 microorganisms-12-01591-f009:**
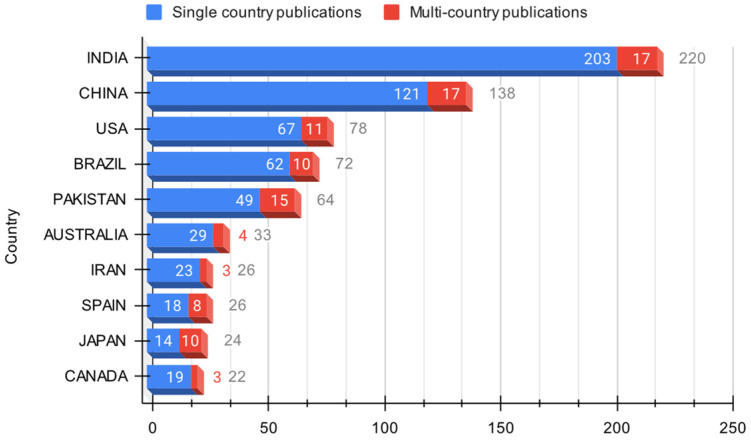
Distribution of collaborations among authors in the domain.

**Figure 10 microorganisms-12-01591-f010:**
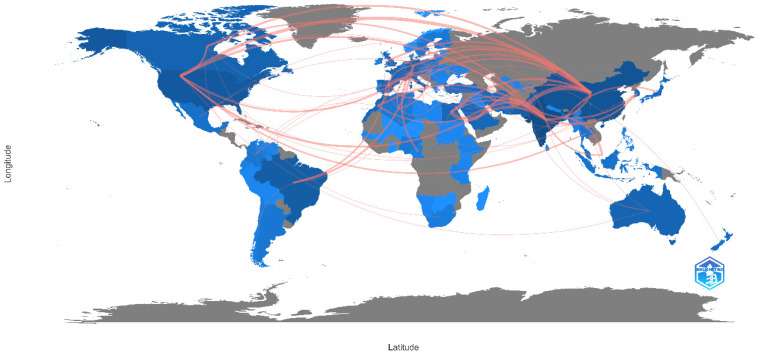
Productive research relationships between different countries in the domain.

**Figure 11 microorganisms-12-01591-f011:**
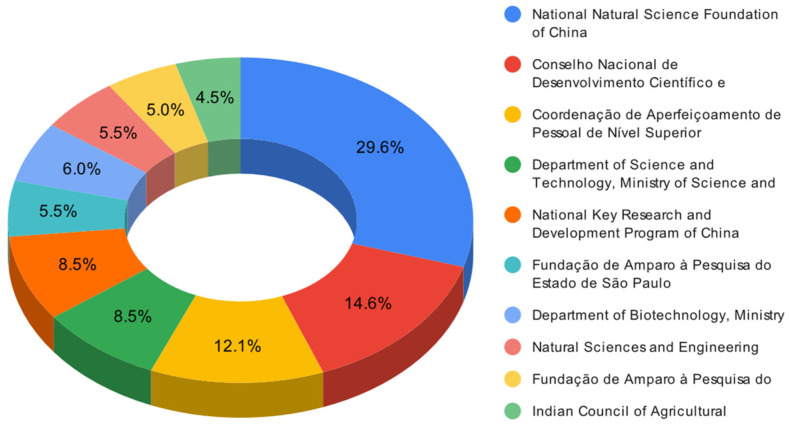
Main research sponsors in the field of solubilizing bacteria as a biotechnological alternative.

**Figure 12 microorganisms-12-01591-f012:**
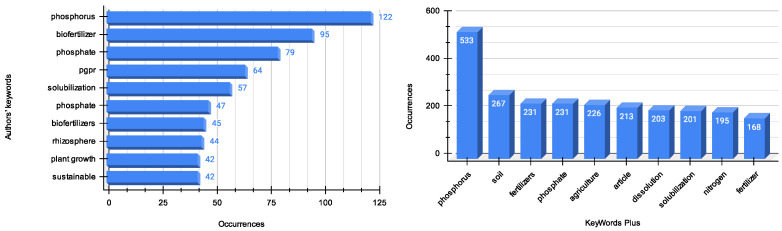
Thematic analysis of the main keywords used in the domain.

**Figure 13 microorganisms-12-01591-f013:**
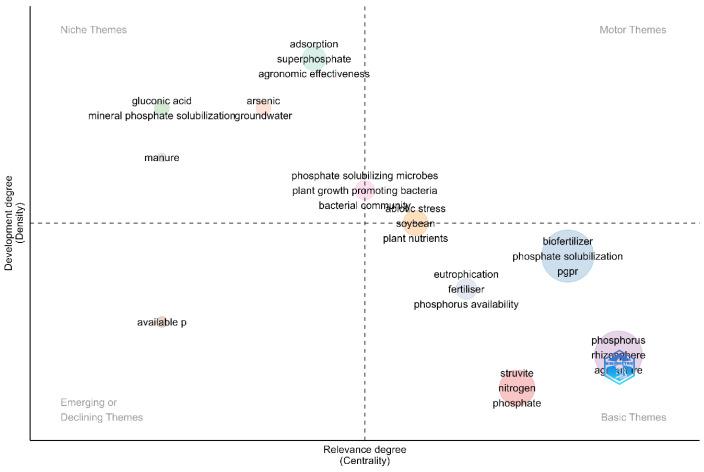
Thematic clustering of the domain for solubilizing bacteria as a biotechnological alternative.

**Figure 14 microorganisms-12-01591-f014:**
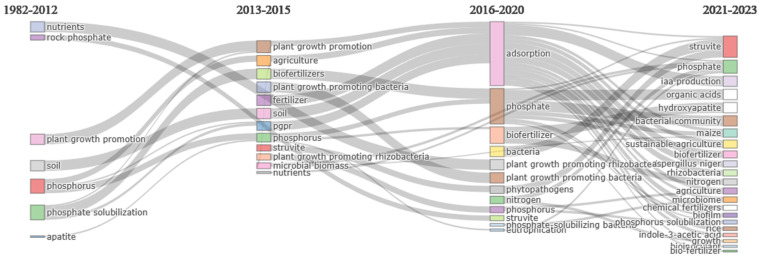
Thematic evolution of the domain of solubilizing bacteria as a biotechnological alternative.

**Figure 15 microorganisms-12-01591-f015:**
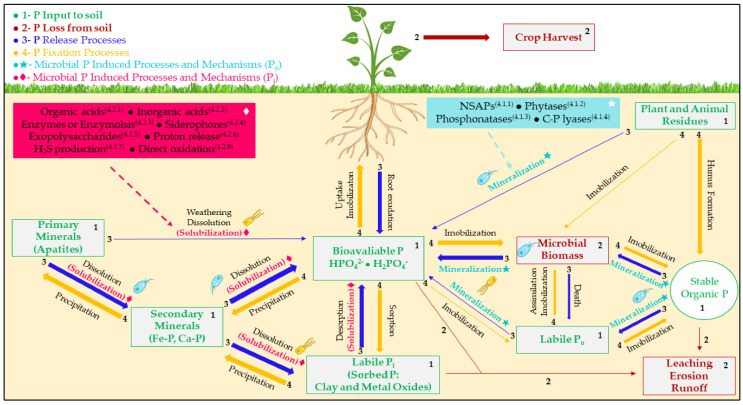
Phosphorus cycle and nutrient mobilization.

**Table 1 microorganisms-12-01591-t001:** Search strategy applied to different scientific databases.

Search Strategy
(Phosphorus OR phosphates OR “phosphate fertilizers”) AND (fertilizers OR “plant food” OR “Soil amendments” OR “Plant nutrients” OR “Soil conditioners” OR Compost OR biofertilizer) AND (agriculture OR farming OR cultivation OR agronomy OR horticulture OR agribusiness OR sustainable) AND (solubilization OR dissolution)

**Table 2 microorganisms-12-01591-t002:** Research questions used in the literature review.

Questions	Motivation
What is the importance of phosphorus in plant physiology or nutrition?	Understand how phosphorus can support the plant’s nutritional process.
What are the sources of phosphorus available in the world?	Understand how phosphorus occurs naturally in the environment and its relationship to the plant environment.
How is phosphorus availability improved using phosphorus-solubilizing microorganisms?	To understand the mechanisms that improve phosphorus availability and new lines of work in this area.
What enzymes or mechanisms are produced by phosphorus-solubilizing microorganisms to enable the plant to absorb phosphorus?	To understand how phosphorus-solubilizing microorganisms improve phosphorus availability for plant uptake.
What are the colonization processes in rhizosphere phosphorus-solubilizing microorganisms?	Understand and investigate how microorganisms in the rhizosphere (the soil region around plant roots) solubilize phosphorus.
What are the prospects for phosphorus-solubilizing microorganisms as a biotechnological solution?	To understand what route this type of biotechnology should take according to the scientific literature.

**Table 3 microorganisms-12-01591-t003:** Outstanding groups of microorganisms for phosphorus solubilization.

Phosphorus-Solubilizing Microorganisms	Crop of Agronomic Interest	Results
*Pseudomonas aeruginosa* [101]	Rice	Increase in plant length, roots, and dry weight by 154.7% and 237.6%.
*Pantoea agglomerans* [102]	Maize	Increase in number of ears/plant and number of seeds/cob by 11.2%, 13.9%, and 11.8%.
*Paenibacillus polymyxa* [103]	Wheat	Increase in plant height, spikelet/spike, and kernels/spike by 16.6%, 16.2%, and 45.6%.
*Pseudomonas* sp. [104]	Chili pepper	Increase in dry weight of aerial part and roots per plant by 11.2% and 7.5%.
*Enterobacter* [105]	Soya	Increase in plant and seed dry weight by 13.8% and 16.1%.
*Aspergillus niger* [106]	Bean	Increase in plant and root length by 50.9% and 27.6%.
*Bacillus subtilis* [107]	Vegetables	Increased available phosphorus in the soil and increased plant growth.
*Azospirillum* [108]	Wheat	Improve biomass and avoid pesticide poisoning.
*Burkholderia cepacian* [109]	Peanut	Increase in the length of stems and roots.
*Funneliformis mosseae* and *Rhizophagus intraradices* [110]	Wheat	Avoid high-stress salt stress and improve leaf area, root volume, N, P, and K/Na.
*Glomus clarum, Gigaspora margarita* and *Glomus etunicatum* [111]	Coffee	Increase in plant height, stem diameter, and yield per hectare of 250 kg.
*Funneliformis mosseae* and *Septoglomus constrictum* [112]	Tomatoes	Increased crop yield and less chemical fertilizer application.
*Rhizophagus irregularis* and *Funneliformis mosseae* [113]	Citrus	Increased plant growth and higher P absorption.

## Data Availability

Using the following link, you can access the material to perform the bibliometric analysis and the literature review, as well as the materials used in this research (https://acortar.link/UP1LXX, accessed on 22 June 2024).

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
