# Peer review of "Use of Phosphorus-Solubilizing Microorganisms as a Biotechnological Alternative: A Review"

_microorganisms, 2024, doi:10.3390/microorganisms12081591_

Round 1

Reviewer 1 Report

Comments and Suggestions for Authors

The review addresses Unlocking Phosphorus with Microbes as a Sustainable Biotechnological Approach.

The review is generally Ok however the following points need to be considered before acceptance

1- Abstract

-Avoid redundancy  in (Despite literature reviews underscoring their contributions in this domain, there remain unexplored territories concerning the accessibility and applicability of these microorganisms.) and {Comprehensive research is necessitated to thoroughly comprehend their potential and efficacious incorporation into sustainable agricultural methodologies)

please consider combine as follows (While literature reviews acknowledge their potential, unexplored territories concerning accessibility, application, and effective integration into sustainable agriculture necessitate further research).

2- Introduction

the Novelty and motivation for this work should be clearly addressed near the end of the introduction.

Special emphasis should be given on how PSMs work to solubilize phosphorus 

4- Please Consider mentioning specific types of struvite precipitation techniques

5- Cite relevant literature appropriately for example in line 108, 550

6- An updated list of PSMs should be given

7- A road map of the potential application of PSMs in sustainable agriculture  should be given.

Comments on the Quality of English Language

Minor editing of English language required

Author Response

Reviewer's comment #1

1- Abstract

-Avoid redundancy  in (Despite literature reviews underscoring their contributions in this domain, there remain unexplored territories concerning the accessibility and applicability of these microorganisms.) and {Comprehensive research is necessitated to thoroughly comprehend their potential and efficacious incorporation into sustainable agricultural methodologies)

please consider combine as follows (While literature reviews acknowledge their potential, unexplored territories concerning accessibility, application, and effective integration into sustainable agriculture necessitate further research).

Response

Accepted: Change made and suggestion taken

Reviewer's comment #2

Introduction

the Novelty and motivation for this work should be clearly addressed near the end of the introduction.

Special emphasis should be given on how PSMs work to solubilize phosphorus Response

Change made and suggestion taken

Response

Accepted: The reviewer's comment was addressed, and a paragraph was added emphasizing how PSMs act to solubilize phosphorus. In addition, the motivation is added

Reviewer's comment #4

Please Consider mentioning specific types of struvite precipitation techniques

Response

Accepted: Added struvite precipitation methods

Reviewer's comment #5

Cite relevant literature appropriately for example in line 108, 550

Response

Accepted: The citations recommended by the reviewer were added, but the complete text was also revised in order to add citations where needed.

Reviewer's comment #6

An updated list of PSMs should be given

Response

Accepted: A table with the main groups of microorganisms for phosphorus solubilization has been added.

Reviewer's comment #7

A road map of the potential application of PSMs in sustainable agriculture  should be given.

Response

Accepted: A new question was added to the review that gives a roadmap that should be followed to improve the use of phosphorus solubilizing bacteria as a b

Reviewer 2 Report

Comments and Suggestions for Authors

1. In section 3.2, a large number of mechanism diagrams need to be added.

2. Add tables to section 3.2 (specific literature examples)

3. Reduce section 3.1. I don't think the extensive analysis and charts in this section are necessary. I suggest deleting section 3.1 directly or reducing it.

4. Add the follow-up research focus on phosphorus solubilizing bacteria at the end of the article.

Author Response

Reviewer's comment #1

  1. In section 3.2, a large number of mechanism diagrams need to be added.

Response

Accepted A table with the main groups of microorganisms for phosphorus solubilization has been added.

Reviewer's comment #2

Add tables to section 3.2 (specific literature examples)

Response

Accepted The table mentioned by the reviewer has been added.

Reviewer's comment #3

  1. Reduce section 3.1. I don't think the extensive analysis and charts in this section are necessary. I suggest deleting section 3.1 directly or reducing it.

Response

Accepted Several subsections of section 3.1 have been reduced without losing scientific integrity.

Reviewer's comment #3

Add the follow-up research focus on phosphorus solubilizing bacteria at the end of the article.

Response

Accepted: The roadmap for this biotechnological alternative has been added.

Reviewer 3 Report

Comments and Suggestions for Authors

The revision of this work has been really very dynamic, since the writing in English has been very understandable, I did not detect any writing problems.

Regarding the subject, however, it should be mentioned that there are multiple documents that relate to the solubilization of phosphate by multiple types of microorganisms.

It is understood that it is a topic highly analyzed by different research groups, who wish to provide new knowledge about the production of fertilizers.

Therefore, this review, being an excellent contribution, diminishes its novelty or impact in the scientific field. I consider that it has excellent arguments in the initial description of the problem, but its contribution tends to be diluted among other works developed.

The argumentation and abundance of data is excellent, the figures presented are very illustrative, which makes it a valuable work.

The references used are abundant, appropriately because within the text the way of choosing the different works chosen for the review is described in a didactic manner.

This way of searching for information can be a very favorable point for this review.

No signs of plagiarism were detected, nor were there any inappropriate self-citations.

I recommend publishing the work without further revisions.

Author Response

Response

Accepted: Thank you for your thoughtful and detailed review of our work. We appreciate the time and effort you have put into providing such comprehensive feedback. We add a paragraph to make the novelty of the article clearer

Reviewer 4 Report

Comments and Suggestions for Authors

Dear authors, many thanks for the opportunity to review the well written review entitled " Use of Phosphorus Solubilizing Microorganisms as a Biotechnological Alternative: A Review”. It is a hot topic and the authors did a lot of work. The review contents are well organized. I believe that this review is worth publishing in “Microorganisms”. However, not all content of your review is described in an easy-to-understand way, and it can be improved. I suggest major revision for this review. Below are my questions, comments, and suggestions.

L12: Rephrase.

L31: Try to focus on what the readers will find new in this review.

L64: solubilizing phosphate

L66: What are the harmful effects of P-chemical fertilizers on environmental pollution? Please clarify? 

L844: Revise I think the authors mean CFU (colony forming unit).

L848-851: Why?

L863: Italicize all scientific names in the whole manuscript.

L818-915: It would be of great importance if the authors included the role of Arbuscular mycorrhizal (AM) symbioses in improving the efficiency of soil phosphorus (P). I believe that AMF are soil microorganisms that play a critical role in this issue.

L984-1083: What about mycorrhizal colonization? It is very important for P availability for plants.

Kind Regards.

Comments on the Quality of English Language

Minor editing of English language required

Author Response

Reviewer's comment #3

L12: Rephrase.

L31: Try to focus on what the readers will find new in this review.

L64: solubilizing phosphate

L66: What are the harmful effects of P-chemical fertilizers on environmental pollution? Please clarify?

L844: Revise I think the authors mean CFU (colony forming unit).

L848-851: Why?

L863: Italicize all scientific names in the whole manuscript.

Response

Accepted: Fixed

Reviewer's comment #3

L818-915: It would be of great importance if the authors included the role of Arbuscular mycorrhizal (AM) symbioses in improving the efficiency of soil phosphorus (P). I believe that AMF are soil microorganisms that play a critical role in this issue.

L984-1083: What about mycorrhizal colonization? It is very important for P availability for plants.

Response

Accepted: Thank you for your comment, the role of arbuscular mycorrhizal (AM) symbiose and L984-1083: What about mycorrhizal colonization? It is very important for P availability for plants. has been added.

Reviewer 5 Report

Comments and Suggestions for Authors

Dear Authors,

Comments for the Manuscript ID: microorganisms-3094998 titled Use of Phosphorus Solubilizing Microorganisms as a Biotechnological Alternative: A Review. The study addressed the significant role of Phosphorus Solubilizing Microorganisms in sustainable agriculture with analysis techniques and statistical methodology. The manuscript was well organized and presented. However, there are some major points that are necessary to consider to improve as below:

1.  All genera and species names need to be in Italics. Please check it in the whole text and in the reference list.

2.  It would be better to have tables for subsections 3.2.3 and 3.2.4 showing the typical Phosphorus Solubilizing Microorganisms and their active mechanism that are recently investigated taking viral roles in plant growth and soil productivity.

Author Response

Reviewer's comment 

  1. All genera and species’ names need to be in Italics. Please check it in the whole text and in the reference list.
  2. It would be better to have tables for subsections 3.2.3 and 3.2.4 showing the typical Phosphorus Solubilizing Microorganisms and their active mechanism that are recently investigated taking viral roles in plant growth and soil productivity.

Response A table with the main groups of microorganisms for phosphorus solubilization has been added. Italian letters have been added to the scientific names of the species.

Round 2

Reviewer 2 Report

Comments and Suggestions for Authors

The author has made certain revisions and improvements, which can be accepted.

Reviewer 4 Report

Comments and Suggestions for Authors

This is the second time I have evaluated this manuscript. The authors addressed all my comments, and the manuscript has been noticeably improved. Many thanks for their contribution.

Comments on the Quality of English Language

The English language is understandable and correct. Only minor editorial and stylistic corrections are required.